# FAST ADVPROP

**Jieru Mei[1], Yucheng Han[2], Yutong Bai[1], Yixiao Zhang[1], Yingwei Li[1], Xianhang Li[3]**
**Alan Yuille[1] & Cihang Xie[3]**

[1]Johns Hopkins University    [2]Nanyang Technological University    [3]UC Santa Cruz

## ABSTRACT

Adversarial Propagation (AdvProp) is an effective way to improve recognition models, leveraging adversarial examples. Nonetheless, AdvProp suffers from the extremely slow training speed, mainly because: a) extra forward and backward passes are required for generating adversarial examples; b) both original samples and their adversarial counterparts are used for training (*i.e.*, $2\times$ data). In this paper, we introduce *Fast AdvProp*, which aggressively revamps AdvProp's costly training components, rendering the method nearly as cheap as the vanilla training. Specifically, our modifications in Fast AdvProp are guided by the hypothesis that disentangled learning with adversarial examples is the key for performance improvements, while other training recipes (*e.g.*, paired clean and adversarial training samples, multi-step adversarial attackers) could be largely simplified.

Our empirical results show that, compared to the vanilla training baseline, Fast AdvProp is able to further model performance on a spectrum of visual benchmarks, *without incurring extra training cost*. Additionally, our ablations find Fast AdvProp scales better if larger models are used, is compatible with existing data augmentation methods (*i.e.*, Mixup and CutMix), and can be easily adapted to other recognition tasks like object detection. The code is available here: https://github.com/meijieru/fast_advprop.

## 1 INTRODUCTION

Deep neural networks are highly successful for visual recognition. As fueled by powerful computational resources and massive amounts of data, deep networks achieve compelling, sometimes even superhuman, performance on a wide range of visual benchmarks. However, when testing out of the box, these exemplary models are usually get criticized for lacking generalization/robustness—an increasing amount of works point out that deep neural networks are brittle at handling out-of-domain situations like natural image corruptions (Hendrycks & Dietterich, 2018), images with shifted styles (Geirhos et al., 2018; Hendrycks et al., 2020), *etc*.

Adversarial propagation (AdvProp) (Xie et al., 2020), which additionally feeds networks with adversarial examples during training, emerged as one of the most effective ways to train not only accurate but also robust deep neural networks. The key in AdvProp is to apply separate batch normalization (BN) layers (Ioffe & Szegedy, 2015) to clean training samples and adversarial training samples, as they come from different underlying distributions. Later works further explore the potential of AdvProp on other learning tasks, including object detection (Chen et al., 2021b; Xu et al., 2021), contrastive learning (Jiang et al., 2020; Ho & Vasconcelos, 2020; Xu & Yang, 2020) and large-batch training (Liu et al., 2022).

However, the benefits brought by AdvProp do not come for "free"—AdvProp introduces a significant amount of additional training cost, which is mainly incurred by generating and augmenting adversarial training samples. For instance, compared to the vanilla training baseline (where only clean images are involved), the default setting in AdvProp (Xie et al., 2020) increase the total computational cost by factor of 7, *i.e.*, 5/7 from generating adversarial examples, 1/7 from training adversarial examples, 1/7 from training clean images. This extremely high training cost not only limits the further explorations of AdvProp on larger networks (Xie et al., 2019; Brock et al., 2021; Dosovitskiy et al., 2020), with larger datasets (Sun et al., 2017; Kuznetsova et al., 2020), and for different learning tasks, but also makes the direct comparisons against other low-cost learning algorithms (Zhang et al., 2018; DeVries & Taylor, 2017; Yun et al., 2019; Cubuk et al., 2019b;a) seemingly unfair.

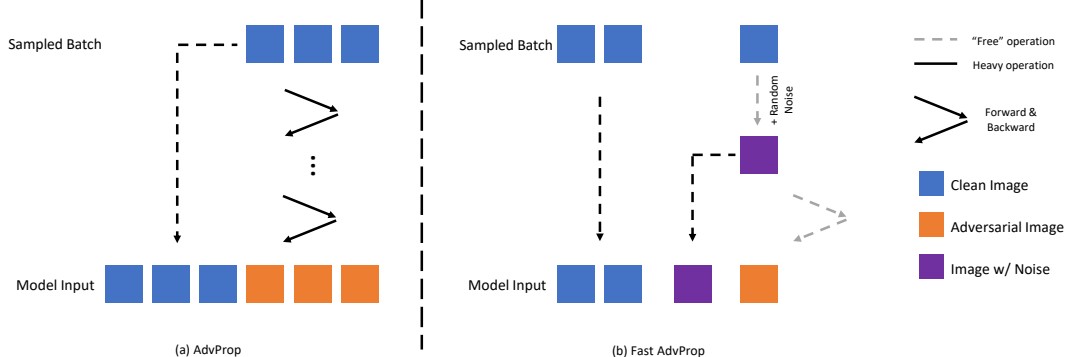

Figure 1: Comparison between AdvProp and Fast AdvProp. (a) AdvProp generates a paired adversarial image (with color *orange*) for each clean image (with color *blue*) in the sampled batch, therefore incurring heavy training cost. Moreover, in addition to clean images, adversarial images are also fed into networks for training, therefore further increasing the total training cost, *i.e.*, 2x data is used here compared to the vanilla training. (b) Different from AdvProp, Fast AdvProp only uses a small portion of the sampled batch to generate adversarial examples. Moreover, during the generation of adversarial images, the gradient calculation of input images and the gradient calculation of network parameters are merged into the same forward and backward pass as in (Shafahi et al., 2019; Zhang et al., 2019), therefore making generating adversarial examples for "free".

In this paper, we present Fast AdvProp, which can run as cheaply as the vanilla training baseline in practice. In particular, noting the most costly training components in AdvProp are (1) *generating adversarial examples* where multiple forward passes and backward passes are additionally required, and (2) *training with both clean samples and their adversarial counterparts* therefore the size of training data gets doubled, Fast AdvProp revamps the original training pipeline as the following:

- Firstly, though both clean training samples and their adversarial counterparts are default components in traditional adversarial training (Goodfellow et al., 2015; Kurakin et al., 2017), we argue such pairing behavior is not a fundamental request by AdvProp. Specifically, in Fast AdvProp, we reposition adversarial examples solely as a *bonus* part for network training, *i.e.*, networks now are expected to train with a mixture of *a large portion of clean images* and *a small portion of adversarial examples*. This adjustment on training data helps lower down training cost. Though the total number of adversarial training examples is reduced, our empirical results verify this strategy is sufficient to let networks gain robust feature representations.

- Secondly, we integrate the recent techniques on accelerating adversarial training (Wong et al., 2020; Zhang et al., 2019; Shafahi et al., 2019) into AdvProp, mainly for reducing the complexity of generating adversarial examples. However, this is non-trivial—naively adopting these fast adversarial training techniques will collapse the training, resulting in suboptimal model performance. We identify such failure is caused by the "label leaking" effect (Kurakin et al., 2017), which largely weakens the regularization power imposed by adversarial training samples. We further note this leakage comes from the intra-batch communication among training samples in the same mini-batch, and resolve it via shuffling BN (He et al., 2020). Additionally, we find a) re-balancing the importance between clean training samples and adversarial training samples and b) synchronizing parameter updating speed are the other two key ingredients for ensuring Fast AdvProp's improvements.

Our empirical results demonstrate that Fast AdvProp can successfully improve recognition models for "free". For instance, without incurring any extra training cost, Fast AdvProp helps ResNet-50 (He et al., 2016) outperforms its vanilla counterpart by 0.3% on ImageNet, 2.1% on ImageNet-C, 1.9% on ImageNet-R and 0.5% on Stylized-ImageNet. Furthermore, such "free lunch" can consistently be observed when Fast AdvProp is applied to networks at different scales, combined with various data augmentation strategies, and adapted to other recognition tasks. By easing the computational barriers, we hope this work can encourage the community to further explore the potential of AdvProp (or adversarial learning in general) on developing better deep learning models.

## 2 RELATED WORK

**Adversarial training.** Adversarial training (Szegedy et al., 2014; Goodfellow et al., 2015), which trains networks with adversarial examples that are generated on the fly, is one of the most effective ways for defending against adversarial attacks. Nonetheless, compared to vanilla training, adversarial training significantly increases the computational overhead, mainly due to the high complexity of generating adversarial examples.

To this end, many efforts have been devoted to accelerating adversarial training. Both (Shafahi et al., 2019) and (Zhang et al., 2019) propose to merge the gradient for adversarial attacks and the gradient for network parameter updates into a single forward and backward pass to reduce computations. Wong *et al*. (Wong et al., 2020) alternatively argue that the cheapest adversarial attacker, Fast Gradient Sign Method (FGSM) (Goodfellow et al., 2015), actually can train robust classifiers, if combined with random initialization. This work is further enhanced by (Andriushchenko & Flammarion, 2020) to explicitly maximizing the gradient alignment inside the perturbation set for enhancing the quality of the FGSM solution. In this work, we aim to integrate these fast adversarial training techniques into AdvProp, for reducing the overhead of generating adversarial training samples.

**Adversarial propagation.** It is generally believed that adversarial training hurts generalization (Raghunathan et al., 2019). Adversarial propagation (AdvProp) (Xie et al., 2020), a special form of adversarial training, challenges this belief by showing training with adversarial examples actually can improve recognition models. The key is to utilize an additional set of batch normalization layers exclusively for the adversarial images (or more importantly, as suggested in (Chen et al., 2021a), by applying a different set of rescaling parameters in batch normalization layers), as they have different underlying distributions to clean examples. Later works further explore the potential of AdvProp on other recognition tasks (Chen et al., 2021b; Xu et al., 2021; Shu et al., 2020; Chen et al., 2021a; Xie & Yuille, 2020; Wang et al., 2020; Gong et al., 2021), under different learning paradigms (Jiang et al., 2020; Ho & Vasconcelos, 2020; Xu & Yang, 2020), with different adversarial data (Merchant et al., 2020; Li et al., 2020; Herrmann et al., 2021), enabling extremely large-batch training (Liu et al., 2022), *etc*. In this paper, rather than furthering performance, we aim to make AdvProp "free".

**Data augmentation.** Data augmentation, which effectively increases the size and the diversity of the training dataset, is crucial for the success of deep neural networks (Krizhevsky et al., 2012; Simonyan & Zisserman, 2015; Szegedy et al., 2015; He et al., 2016). Popular ways for augmenting data include geometric transformations (*e.g*., translation, rotation), color jittering (*e.g*., brightness, contrast), mixing images (Zhang et al., 2018; Yun et al., 2019; DeVries & Taylor, 2017), *etc*.

Training with adversarial examples (Goodfellow et al., 2015; Xie et al., 2020; Chen et al., 2021b) can be regarded as a special way to augment data—different from traditional data augmentation strategies which are usually fixed and model agnostic, the policy of generating adversarial examples is jointly evolved with the model updating throughout the whole training process. This behavior ensures the augmentation policy of adversarial examples stays current and relevant. Nonetheless, a significant drawback of augmenting adversarial examples is that the introduced computational overhead is much more expensive than that of traditional augmentation strategies. We hereby aim to make training with adversarial examples as cheap as other data augmentation strategies.

## 3 FAST ADVPROP

We hereby present Fast AdvProp, which aggressively revamps the costly components in AdvProp. Particularly, our modifications mainly focus on reducing the computational overheads stemmed from adversarial examples, meanwhile (empirically) still attempt to retain the benefits brought by AdvProp.

### 3.1 REVISITING ADVPROP

AdvProp (Xie et al., 2020) demonstrates adversarial examples can improve recognition models. By noticing adversarial images and clean images have different underlying distributions, AdvProp bridges such distribution mismatch by using two BN scheme—the original BN layers are applied exclusively for clean images, and the auxiliary BN layers are applied exclusively for adversarial images. This scheme ensures each BN layer is executed on a single data source (*i.e*., either clean images or adversarial images). More concretely, in each iteration,

Table 1: The performance of vanilla training, AdvProp, same-budget AdvProp, and Fast AdvProp on various datasets. ↑/↓ indicate the higher/lower the better.

| | IMAGENET ↑ | IMAGENET-C ↓ | IMAGENET-R ↑ | S-IMAGENET ↑ | TRAINING BUDGET |
|---|---|---|---|---|---|
| Vanilla Training | 76.2 | 58.5 | 36.3 | 7.8 | 1× |
| PGD-5 AdvProp | 77.0 | 52.0 | 42.3 | 12.0 | 7× |
| 15-epoch PGD-5 AdvProp | 66.8 | 66.2 | 31.7 | 8.6 | 1× |
| PGD-1 AdvProp | 77.5 | 54.3 | 39.5 | 8.6 | 3× |
| 35-epoch PGD-1 AdvProp | 73.9 | 59.6 | 36.5 | 9.3 | 1× |
| Fast-AdvProp (ours) | 76.5 | 56.4 | 38.2 | 8.3 | 1× |

- **Step 1:** After randomly sampling a subset of the training data, AdvProp applies the adversarial attack to these clean images for generating the corresponding adversarial images, using the auxiliary BN layers.

- **Step 2:** The clean images and their adversarial counterparts are then fed into the network as a pair. Specifically, the original BN layers are applied exclusively on the clean images, and the auxiliary BN layers are applied exclusively on the adversarial images.

- **Step 3:** The loss from adversarial images and clean images are jointly optimized for updating network parameters.

As shown in (Xie et al., 2020), AdvProp substantially improves both the clean images accuracy, as well as the model robustness. We confirm it in our re-implementation—as shown in the second row of Table 1, AdvProp, using PGD-5 attacker, helps ResNet-50 beats its vanilla counterpart by 0.8% on ImageNet (Russakovsky et al., 2015), 6.5% on ImageNet-C (Hendrycks & Dietterich, 2018), 6.0% on ImageNet-R (Hendrycks et al., 2020) and 4.2% on Stylized-ImageNet (Geirhos et al., 2018).

But meanwhile, we note AdvProp significantly increases the training cost. For example, our AdvProp re-implementation requires 7× more forward and backward passes than the vanilla baseline. Such heavy training cost not only limits the broader exploration with AdvProp, but also makes the comparisons to other learning strategies (which are usually "free", *e.g.*, (Yun et al., 2019; Zhang et al., 2018; Cubuk et al., 2019a)) seemly unfair.

To reduce the computational cost, we first give a naive attempt to simplify AdvProp's training pipeline. Specifically, given PGD-5 AdvProp here is 7× more expensive than the vanilla baseline, we directly cut its total training epochs by a factor of 7 (*i.e.*, from 105 epochs to 15 epochs). As shown in the third row of Table 1, this *15-epoch PGD-5 AdvProp* severely degrades the original AdvProp's performance (*i.e.*, 66.8% *vs*. 77.0% on ImageNet), even making the resulted model attains much lower performance than the vanilla training baseline. Moreover, we verify that applying the cheapest PGD-1 training (*i.e.*, FGSM + random initialization as in (Wong et al., 2020)) to AdvProp still leads to inferior performance. These results demonstrate that the task of accelerating AdvProp is non-trivial, therefore motivate us to explore more sophisticated solutions next.

## 3.2 LIGHTENING ADVPROP

We hereby carefully diagnose the design choices of AdvProp, aiming to simplify/purge its costly training components. Note that in our ablations, we always keep the disentangled learning behavior with adversarial training samples unchanged (*i.e.*, keep separate BN layers for adversarial samples and clean samples), as we assume this is the key for gaining robust features from adversarial examples.

### 3.2.1 UNPAIRING TRAINING SAMPLES IN ADVPROP

Let's consider the training cost of one epoch. We denote the cost of a single forward and backward pass for one image as $1$ [1], and the dataset size as $N$. Then the cost of vanilla training for one epoch is

$$cost(\text{Vanilla}) = N. \tag{1}$$

---

[1]During a standard training step, we compute the gradient for all parameters used in the forward process; while for attacking the network, we only compute the gradient with respect to the input images (therefore less computations). Nonetheless, we ignore such differences for simplification in cost calculation.

Table 2: The performance of AdvProp with different settings. +ADVPROP denotes the original AdvProp. +1 ITER denotes the AdvProp with PGD-1 attacker. +DECOUPLED denotes the decoupled training where only a small portion (*e.g.*, 20% here) of training images are used for generating adversarial examples. The last column reports the corresponding training budget in one epoch.

| +ADVPROP | +1 ITER | +DECOUPLED | IMAGENET ↑ | IMAGENET-C ↓ | TRAINING BUDGET |
|----------|---------|------------|------------|--------------|-----------------|
|          |         |            | 76.2       | 58.5         | N               |
| ✓        |         |            | 77.0       | 51.9         | 7N              |
| ✓        | ✓       |            | 77.5       | 54.3         | 3N              |
| ✓        | ✓       | ✓          | 76.6       | 56.8         | 1.2N            |

Similarly, the cost of AdvProp using PGD-$K$ attack (Madry et al., 2018) for one epoch is:

$$cost(\text{AdvProp}) = N + K * N + N = (K + 2) \times N, \qquad (2)$$

where the first part (*i.e.*, $N$) refers to the training cost of clean samples, the second part (*i.e.*, $K * N$) refers to the cost of generating adversarial examples, and the third part (*i.e.*, $N$) refers to the training cost of adversarial examples. Note AdvProp by default use $K = 5$ (Xie et al., 2020), therefore increasing the training cost by a factor of 7 compared to the vanilla training baseline. This high training cost makes further scaling AdvProp to the *large-computing* settings (Xie et al., 2019; Mahajan et al., 2018; Dosovitskiy et al., 2020) challenging.

**PGD-1 attack.** Firstly, rather than using $K = 5$, we use $K = 1$ as the default setting to reduce the training cost from $7N$ to $3N$. Note this change simplify the PGD attacker to *the FGSM attacker (Goodfellow et al., 2015) with random noise initialization*. As shown in Table 2, compared to the default AdvProp, this simplification increases the top-1 accuracy by 0.5% on ImageNet, but at the cost of sacrificing the robustness on ImageNet-C (*i.e.*, 2.4 higher mCE).

**Decoupled training.** Secondly, AdvProp implicitly introduces a constraint that, for each clean image, we should generate a paired adversarial image for jointly training. Though such paired training behavior is popular and standard in adversarial training (Goodfellow et al., 2015), interestingly, we empirically find this is not necessarily needed, *i.e.*, models can still be benefited from training with non-paired clean images and adversarial images. Moreover, if we choose to break such pairing behavior, the training cost will be reduced to:

$$cost(\text{Fast AdvProp}) = p_{clean} * N + p_{adv} * (K + 1) * N \qquad (3)$$

where $p_{clean}$ is the percentages of training images used as clean examples, and $p_{adv}$ is the percentages of training images used as adversarial examples. Note AdvProp by default sets $p_{clean} = p_{adv} = 1$. While for Fast AdvProp, we exclusively set $p_{adv}$ of the training samples for generating adversarial examples and keep the rest as clean examples. The training cost now is

$$cost(\text{Fast AdvProp}) = (1 - p_{adv}) * N + p_{adv} * (K + 1) * N = (p_{adv} * K + 1) * N \qquad (4)$$

As shown in Table 2, by setting $p_{adv} = 0.2$ and apply PGD-1 attacker, our Fast AdvProp not only largely reduces the training cost to $1.2N$, but also beats the vanilla training baseline by 0.4% top-1 accuracy on ImageNet and by 1.7 mCE on ImageNet-C.

### 3.2.2 INCORPORATING FREE ADVERSARIAL TRAINING TECHNIQUES

Though decoupled training (using Equation 4) enables AdvProp to use the same number of training images as the vanilla setting, the resulted strategy still incurs extra training cost. This is because additional forward-backward passes are exclusively reserved for generating adversarial examples.

Inspired by the recent works on accelerating adversarial training by recycling the gradient (Shafahi et al., 2019; Zhang et al., 2019), we aim to improve the gradient utilization in generating adversarial examples, *i.e.*, such gradients will also be used for updating network parameters. In this way, we are able to let networks additionally train with the samples that are in the intermediate state of generating adversarial examples (*e.g.*, clean image with random noise), and more importantly, for "free". Note that, following Shafahi et al. (2019), in order to re-using gradient, *we need to switch the attack mode from targeted attack to untargeted attack in Fast AdvProp*.

---

**Algorithm 1:** Pseudo code of Fast AdvProp for $T$ epochs, given some radius $\epsilon$, importance re-weight parameter $\beta$, learning rate $\gamma$, and ratio of adversarial examples $p_{adv}$.

---

**Data:** A set of clean images with labels;
**Result:** Network parameter $\theta$
**for** $epoch = 1, \ldots, T/(p_{adv}+1)$ **do**

    Sample a clean image mini-batch $X$ with label $Y$
    Split $X$ into $X_1, X_2$ with the ratio $1 - p_{adv}$ and $p_{adv}$, respectively
    Generate the adversarial examples $X_{adv}$ using $X_2$
        $\delta \sim \mathcal{U}(-\epsilon, \epsilon)$
        $g_\delta \leftarrow \nabla_\delta l(x + \delta, y, \theta)$
        $\delta' \leftarrow \delta + \epsilon \cdot sign(g_\delta)$
        $X_{adv} = X_2 + clip(\delta', -\epsilon, \epsilon)$
    Calculate gradients for $X_1, X_2, X_{adv}$, note $g_\theta^{noise}$ and $g_\delta$ could be compute simultaneously
        $g_\theta^{clean} \leftarrow \mathbb{E}_{x \in X_1}[\nabla_\theta l(x, y, \theta)]$
        $g_\theta^{noise} \leftarrow \mathbb{E}_{x \in X_2}[\nabla_\theta l(x + \delta, y, \theta)]$
        $g_\theta^{adv} \leftarrow \mathbb{E}_{x \in X_{adv}}[\nabla_\theta l(x, y, \theta)]$
    Re-balance the gradients
        $g_\theta \leftarrow g_\theta^{clean} + \beta \cdot g_\theta^{noise} + \beta \cdot g_\theta^{adv}$
    Update $\theta$ using gradient descent
        $\theta \leftarrow \theta - \gamma g_\theta$

**end**

---

As analyzed by Equation 4, the training cost of Fast AdvProp now is $(p_{adv} * K + 1) * N$ for one epoch. To let the training cost of Fast AdvProp exactly match the training cost of the vanilla setting, we further calibrate the training by reducing the total number of epochs by a factor of $(p_{adv} * K + 1)$.

Following the design principles above, interestingly, we found naively recycling the gradients makes the training of Fast AdvProp unstable. Next, as summarized in Algorithm 1, we stabilize the network training using the following techniques:

**Bridging the gap between the running/batch statistics in BN.** Adversarial attackers by default attack models using their testing mode, *e.g.*, the running statistics is applied in BN. Nonetheless, we observe unstable training with NaN loss if we reuse the gradient and using BN's running statistics at the same time. We conjecture this comes from the inconsistent gradient paths—we use the batch statistics for training with the clean examples and adversarial examples, but use the running statistics for generating adversarial examples. To remove that inconsistency, we now use the batch statistics for generating adversarial examples.

**Shuffling BN for dealing with information leakage** When attacking using batch statistics, we observe the training accuracy is unreasonably high, as shown in Figure 2. The network could classify images by "cheating" since we use the same batch for attacking and training, and the intra-batch information exchange introduced by BN leaks information. We use shuffling BN (He et al., 2020) to resolve this problem. Before feeding the adversarial images on each GPU into the network, we shuffle the generated adversarial examples across the multiple GPUs. This ensures the batch statistics for the images with random noise and the adversarial images come from different subsets, therefore preventing information leakage.

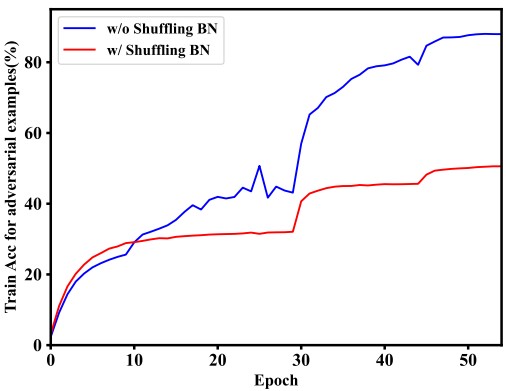

Figure 2: Illustration of the information leakage.

**Re-balancing training samples.** By reusing the gradient, the images used for adversarial attack involve two forward and backward passes (*i.e.*, one for generating adversarial examples, and one for training with adversarial examples), while the clean images involve only one forward and backward pass. With the intuition that each image within the batch should have the same importance, we propose to halve the importance of the images used for adversarial attack. In another perspective, the adversarial images and their intermediate products (*i.e.*, the images with random noise) are two kinds of augmentations on the original images. Based on the argument in (Hoffer et al., 2020) that the overall importance of an image should be the same w/ or w/o the repeated augmentations, we therefore need to adjust the importance of the images with random noise and the importance of the adversarial examples accordingly (*i.e.*, setting $\beta = 0.5$ in Algorithm 1).

**Synchronizing parameter updating speed.** The decoupled training strategy enables us to substantially reduce the training cost. Nonetheless, it causes problems when combining with the auxiliary BNs scheme. Specifically, the parameters of original BNs only receive the gradients from clean images, the parameters of auxiliary BNs only receive the gradients from images with random noise and adversarial examples, while the parameters of the shared layers receive gradients from all examples. The ratio between the gradient magnitude of shared layers/original BNs/auxiliary BNs is $1 : (1 - p_{adv}) : p_{adv}$. The inconsistent updating speed of network parameters harms the performance. To solve this problem, we re-scale the gradient to ensure the similar parameter updating speed.

## 4 EXPERIMENTS

### 4.1 EXPERIMENTS SETUP

**Dataset.** We evaluate model performance on ImageNet classification, and the robustness on different specialized benchmarks including ImageNet-C, ImageNet-R, and Stylized ImageNet. ImageNet dataset contains 1.2 million training images and 50000 images for validation of 1000 classes. ImageNet-C measures the network robustness on 15 common corruption types, each with 5 severity. ImageNet-R contains stylish renditions like cartoons, art, and sketches of 200 ImageNet classes resulting in 30000 images. Stylized-ImageNet dataset keeps the global shape information while removing the local texture information using the AdaIN style transfer (Huang & Belongie, 2017).

**Implementation Details** We use the renowned ResNet family (He et al., 2016) as our default architectures. We use a SGD optimizer with momentum 0.9 and train for 105 epochs. The learning rate starts from 0.1 and decays at 30, 60, 90, 100 epochs by 0.1. We use a batch size of 64 per GPU for vanilla training. For decoupled training setting, we use a batch size of $64/(1 - p_{adv})$ per GPU, keeping the same 64 batch size per GPU for the original BNs. $p_{adv}$ is set to 0.2 if not specified.

For a strictly fair comparison, we scale the total epochs and decay epochs by the relative training cost to vanilla training (*i.e.*, $p_{adv} * K + 1$ in Fast AdvProp, $K + 2$ in AdvProp) in the same budget setting. To generate adversarial images, we using the PGD attacker with random initialization. We attack for one step ($K = 1$) and set the perturbation size to 1.0. As discussed in Section 3.2.2, to ensure the same importance of each example within a batch, we set $\beta = 0.5$ for halving the importance of the images with random noise and the adversarial images. Additionally, we re-scale the gradient to achieve the $1 : 1 : 1$ ratio for ensuring similar updating speed of all parameters.

### 4.2 MAIN RESULTS

Table 1 compares our method with AdvProp and the vanilla training using ResNet-50. Firstly, our re-implementation of AdvProp achieves 77.0% top-1 accuracy on ImageNet, 0.8% higher than the vanilla training baseline. We also evaluate robustness generalization on ImageNet-C, ImageNet-R, Stylized-ImageNet; AdvProp also substantially outperforms the vanilla training baseline here. However, the comparison is unfair as AdvProp using $7\times$ training budget. In the fair comparison setting, the performance of 15-epoch AdvProp (dividing the total epochs by 7) degrades significantly, *i.e.*, its ImageNet accuracy is only 66.8%, which is 9.4% lower than the vanilla training baseline.

On the contrary, using exactly the same training budget as the vanilla setting, Fast AdvProp achieves 76.5% accuracy on ImageNet, which is 0.3% higher than the vanilla training baseline and significantly higher than the 15-epoch AdvProp. As shown in Table 1, our Fast AdvProp also shows stronger robustness than both the vanilla training baseline and the 15-epochs AdvProp.

Table 3: The ImageNet accuracy and robustness generalization of vanilla training and Fast AdvProp.

| | IMAGENET ↑ | IMAGENET-C ↓ | IMAGENET-R ↑ | S-IMAGENET ↑ |
|---|---|---|---|---|
| ResNet-50 | 76.2 | 58.5 | 36.3 | 7.8 |
| + 15-epoch PGD-5 AdvProp | 66.8 | 66.2 | 31.7 | 8.6 |
| + Fast AdvProp | 76.5 | 56.4 | 38.2 | 8.3 |
| ResNet-101 | 77.8 | 54.7 | 40.1 | 9.1 |
| + 15-epoch PGD-5 AdvProp | 68.6 | 61.8 | 34.0 | 12.3 |
| + Fast AdvProp | 77.8 | 51.7 | 41.0 | 10.8 |
| ResNet-152 | 78.3 | 52.3 | 40.5 | 10.5 |
| + 15-epoch PGD-5 AdvProp | 69.0 | 59.6 | 36.2 | 13.2 |
| + Fast AdvProp | 78.6 | 49.8 | 42.0 | 12.4 |

Additionally, our method scales better with larger networks. As shown in Table 3, Fast AdvProp helps the large ResNet-152 achieve 78.6% top-1 accuracy on ImageNet, 49.8 mCE on ImageNet-C, 42.0% top-1 accuracy on ImageNet-R and 12.4% top-1 accuracy on Stylized-ImageNet, beating its vanilla counterpart by 0.3% on ImageNet, 2.5% on ImageNet-C, 1.5% on ImageNet-R and 1.9% on Stylized-ImageNet, respectively. This observation also holds when comparing Fast AdvProp to the 15-epoch AdvProp baseline (which has the same training cost as ours). Table 3 shows that, for all ResNet models, Fast AdvProp attains much higher performance than the 15-epoch AdvProp baseline on ImageNet, ImageNet-C and ImageNet-R. The only exception is Stylized-ImageNet, where 15-epoch AdvProp always attains the best performance. We conjecture this is mainly due to a larger percentage of adversarial data is used during training, therefore letting the learned feature representation be biased towards shape cues (Geirhos et al., 2018; Zhang & Zhu, 2019).

### 4.3 ABLATION STUDY

**The importance of decoupled training.** In Table 4, we take a close look at the effect of $p_{adv}$. We can draw a clear conclusion that the larger the $p_{adv}$, the more inferior top-1 accuracy we achieved. Specifically, using 50% images as adversarial examples only gets 75.4% accuracy, which is 0.9% lower than the vanilla training. This comes from the fact that as $p_{adv}$ increases, the training cost in one epoch increases (calculated using Equation 4), therefore the total training epoch needs to be reduced for keeping the training cost unchanged. In addition, reducing $p_{adv}$ from 0.20 to 0.11 does not further decrease the model accuracy, suggesting its value is fairly robust in a certain range. If we further decrease $p_{adv}$ to 0, then our method degenerates to the vanilla training baseline.

The decoupled training strategy enables us to train the networks with a mixture of a large portion of clean images and a small portion of adversarial examples, which helps the network go through the dataset with enough epochs. Based on Table 4, we choose $p_{adv} = 0.2$ as our default setup.

Table 4: Ablation study on the influence of the percentage of the adversarial images.

| $p_{adv}$ | IMAGENET ↑ | EPOCHS |
|---|---|---|
| 0.00 | 76.2 | 105 |
| 0.11 | 76.5 | 94 |
| 0.20 | 76.5 | 87 |
| 0.33 | 76.0 | 79 |
| 0.50 | 75.4 | 70 |

**The importance of example re-balancing and updating speed synchronization.** In the default setting, the weight ratio between the clean/noise/adversarial examples is $1 : 1 : 1$ without adjustment. With $p_{adv} = 0.2$, the ratio of updating speed between the shared/clean only/adversarial only parameters therefore is $1 : 0.8 : 0.2$. We cannot observe any performance gain in this setting as shown in Table 5, *i.e.*, its ImageNet accuracy is 76.2%, almost the same as the vanilla training.

We find it is important to simultaneously adopt those two strategies, which achieves 76.53% ImageNet accuracy. Ignoring the re-balancing for the examples hurts the accuracy by 0.18%; this may because some examples now are more important than the others, therefore violating the assumption that the overall importance for each example should be the same (Hoffer et al., 2020). Removing the updating speed synchronization hurts the performance as well, leading to 76.27% ImageNet accuracy, which is 0.26% lower than the setting with consistent parameter updating speed.

Table 5: Ablation study of the effects on re-balancing samples and synchronizing updating speed.

| SYNCHRONIZING THE UPDATE SPEED | SAME IMPORTANCE | IMAGENET ↑ |
|:---:|:---:|:---:|
| | | 76.20 |
| | ✓ | 76.27 |
| ✓ | | 76.35 |
| ✓ | ✓ | **76.53** |

**Information leakage problem.** We find that shuffling BN is important for gradient reusing. Without shuffling BN, the training accuracy on the adversarial images reaches 88.0%, even higher than the training accuracy on the clean images (73.3%). This observation is counter-intuitive, as the adversarial images are much more difficult than the clean images. The shuffling BN technique resolves this problem effectively—we observe a smooth and reasonable training curve in Figure 2. In addition, the validation accuracy boosts from 74.5% to 75.1%.

**Combining with other data augmentations.** AdvProp could be viewed as a data augmentation method from the perspective of increasing the size and diversity of the dataset using adversarial examples. We next combine Fast AdvProp with common data augmentation methods including Mixup (Zhang et al., 2018) and CutMix (Yun et al., 2019). Specifically, compared with the vanilla training baseline, Mixup and CutMix require extra training epochs to fully cope with their strong regularization. For Mixup, we train 180 epochs and decay the learning rate by 0.1 every 60 epochs. For CutMix, we train for 210 epochs and decay the learning rate at 75, 150 and 180, respectively. To properly integrate them with Fast AdvProp, we apply the extra data augmentations on clean images only. Note that the training cost is the same w/ or w/o our method.

Fast AdvProp achieves comparable performance on ImageNet but much better robustness. As shown in Table 6, when combining with CutMix, our method beats the baseline by 2.1% on ImageNet-C, 1.3% on ImageNet-R, 2.5% on Stylized-ImageNet. These results suggest Fast AdvProp is compatible with existing data augmentations for furthering the network performance and robustness.

Table 6: Robustness when combining our Fast AdvProp with existing data augmentation methods.

| | IMAGENET-C ↓ | IMAGENET-R ↑ | S-IMAGENET ↑ |
|:---|:---:|:---:|:---:|
| CutMix | 58.9 | 35.2 | 5.5 |
| + Fast AdvProp | 55.0 | 36.5 | 7.0 |
| MixUp | 53.4 | 41.0 | 10.0 |
| + Fast AdvProp | 52.2 | 41.7 | 10.9 |

**Object detection results.** We implement Fast AdvProp on object detection and evaluate it on COCO dataset (Lin et al., 2014). We adopt RetinaNet (Lin et al., 2020) as the detection framework without freezing the BN's running statistics. We train 24 epochs for the baseline and 20 epochs for Fast AdvProp. We benchmark both the standard performance on COCO and the robustness on COCO-C with 15 common corruption types (Michaelis et al., 2019). As shown in Table 7, compared to the baseline, our method achieves the similar performance on COCO detection, but much better robustness (i.e., +1.4%) in COCO-C.

Table 7: Object detection mAP(%) and robustness measurements on COCO and COCO-C.

| | COCO (mAP clean) | COCO-C (mAP corr.) |
|:---|:---:|:---:|
| Vanilla Training | 35.8 | 17.6 |
| + Fast-AdvProp | 35.8 | 19.0 |

## 5 CONCLUSION

AdvProp is an effective method to get accurate and robust networks. However, it suffers from extremely high training costs. We propose the decoupled training where networks are expected to train with only a small portion of adversarial examples but a large portion of clean images, therefore reducing the training cost significantly. In addition, we succeed in reusing the gradient when generating adversarial examples. Empirically, our Fast AdvProp effectively and efficiently helps network gain better performance without incurring extra training cost. We believe our method will speed up the iteration of future works and accelerating the research on adversarial learning.

ACKNOWLEDGEMENTS

This work was supported by ONR N00014-21-1-2812. Cihang Xie was supported by a gift grant from Open Philanthropy.

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

# A APPENDIX

## A.1 DO WE NEED RANDOM INITIALIZATION IN PGD ATTACK?

Here we focus on ablating the effects of the random initialization in PGD attacker on Fast AdvProp. Note that removing the random initialization simplifies the PGD-1 attacker to the FGSM attacker. Though, as discussed in (Wong et al., 2020), the random initialization is the key step for enabling successful adversarial training, we observe Fast AdvProp behaves robustly under such situation—it achieves 76.46% top-1 ImageNet accuracy (using FGSM), which closely matches the situation when random initialization is applied (*i.e.*, 76.53% top-1 ImageNet accuracy, using PGD-1).

