# OpenReview forum: "Fast AdvProp"
_ICLR.cc/2022/Conference — ICLR 2022 Poster_

### Official Review · Reviewer_QrYp · 2021-10-25

**Correctness:** 3
**Technical Novelty And Significance:** 2
**Empirical Novelty And Significance:** 2
**Recommendation:** 5
**Confidence:** 4

**Main Review:**

The paper actually has resolved one of the main issues of AdvProp, making it more efficient. The empirical study shows very promising results. However, I still have some concerns about the work:

1. The novelty and technique strength is limited. The way to combine clean and adversarial batches is empirical. The trick used for fast adversarial training has been explored by many other previous works.
2. The empirical study only shows the effectiveness of fast AdvProp on the ResNet family, but not EfficientNets. In addition, on the detection task, the performance improvement is not that significant.
3. Some claims and empirical supports of the paper are confusing. For example, [chen et al. 2021a] uses an adversarial feature augmentation instead of doing something like AdvProp to generate real adversarial samples. For me, AdvProp has more advantages to improve the clean accuracy. But the paper shows some experiments on the robustness side.

A few notice about more technical details regarding AdvProp: 1. for the TensorFlow version, PGD-5 is better than PGD-1. But the Pytorch version [a] has a reversed conclusion; 2. the two BN trick was analyzed by the paper [b], I am not sure if the author has a similar observation on that.

a.https://github.com/tingxueronghua/pytorch-classification-advprop
b.Adversarial Masking: Towards Understanding Robustness Trade-off for Generalization

**Summary Of The Paper:**

This paper proposed an improved version of AdvProp, to speed up the training process. Specifically, it proposes to use less adversarial examples in a mixed batch and utilizes a list of recent fast adversarial training techniques. The experiments have been conducted on ImageNet sets with different backbones, demonstrating the effectiveness of the proposed fast model.

**Summary Of The Review:**

Overall, I think the paper is interesting but has some technical details to be addressed. In addition, the novelty of the paper is increamental.

---

> ### Author Response · Authors · 2021-11-22
> **Reply to Reviewer QrYp**
>
> We first thank the reviewer for the detailed comments and the appreciation of our work. We address the concerns below:
>
>
> **Q1: Our method is a collection of existing methods**
>
> A1: Thanks for raising this concern. As we stated in the **general response**, enabling these existing strategies to improve model performance is challenging, and requires a significant amount of non-trivial diagnosis and technical contributions. We will include these clarifications in the paper.
>
>
> **Q2: Results on EfficientNets**
>
> A2: In practice, EfficientNets run much slower than ResNets, mainly due to the usage of depthwise convolution. For examples, compared to ResNet-50 (#Params=26M), the much smaller EfficientNet-B0 (#Params=5.3M) shows the similar training speed. This observation is also empirically validated in [1] (see ResNet-50’s training time in Table 3 and EfficientNet-B0’s training time in Table 4). In addition, EfficientNet by default requires 350 training epochs, which is much more expensive than ResNet’s 105 training epoch setup. Therefore, given the heavy computational cost of EfficientNets, we choose to build our results mainly based on ResNets in this paper. Nonetheless, we believe our Fast AdvProp should be able to improve EfficientNet as well.
>
>
> **Q3: Improvements on detection task are not significant.**
>
> A3: As shown in [2] (see Table 6), the full version of AdvProp is able to help the EfficientDet family gain +0.8 ~ +2.3 mAP improvement on COCO-C. Specifically, for small EfficientDet models like EfficientDet-D0 and EfficientDet-D1, the improvements are +0.8% and +1.7%, respectively. Therefore we believe +1.4% mAP on COCO-C with the small ResNet-50 is a reasonable improvement.
>
>
> **Q4: Why have robustness results**
>
> A4: As shown in the original AdvProp paper, AdvProp successfully improves both the clean image accuracy and robustness on out-of-distribution samples. Therefore we follow this evaluation protocol to report the model's clean accuracy and robustness.
>
>
> **Q5: PyTorch implementation v.s. Tensorflow implement regarding PGD-5 training and PGD-1 training**
>
> A5: Firstly, we would like to point out that the results from these two implementations do not contradict each other. As shown in Table 2 of the original AdvProp paper, PGD-1 shows better performance than PGD-5 for small models (e.g., EfficientNet-B0), and PGD-1 shows worse performance than PGD-5 for large models (e.g., EfficientNet-B7). This is mainly due to models at different scales requiring different regularization strength---small models require weak regualizations (like PGD-1) and big models require strong regularization (like PGD-5). This is also discussed in Section 5.2 “Ablation on Adversarial Attacker Strength” in the original AdvProp paper. Therefore, given ResNet-50 is a small model, it is expected that PGD-1 outperforms PGD-5.
>
>
> **Q6: Related work [3]**
>
> A6: Thanks for bringing in this related work. We believe learning an adaptive feature masking for different perturbation strengths could help design better Fast AdvProp. We will discuss this related work in the paper.
>
>
> [1] Radosavovic, Ilija, et al. "Designing network design spaces." Proceedings of the IEEE/CVF Conference on Computer Vision and Pattern Recognition. 2020.
>
> [2] Chen, Xiangning, et al. "Robust and accurate object detection via adversarial learning." Proceedings of the IEEE/CVF Conference on Computer Vision and Pattern Recognition. 2021.
>
> [3] Cheng, Minhao, et al. "Adversarial Masking: Towards Understanding Robustness Trade-off for Generalization." (2020).

---

### Official Review · Reviewer_7oqV · 2021-10-29

**Correctness:** 3
**Technical Novelty And Significance:** 2
**Empirical Novelty And Significance:** 2
**Recommendation:** 5
**Confidence:** 4

**Main Review:**

Strengths of this paper:

Highlighted several "tricks" to save on computational cost for AdvProp. I recognized that the discovery of these tricks and that they work to a certain extent is non-trivial work and appreciate those insights. Namely, they are (a) decoupling does not hurt as much, (b) 1-step adversarial does not cause as much degradation as previously thought, (c) recycling gradient works better with (i) non-targeted attack, (ii) using batch statistic for adversarial image generation, (iii) using shuffle BN to avoid label leaks, (iv) rebalancing examples and (v) synchronizing the update speed of parameters.

The paper is also written very clearly and the motivation is well communicated.

Weaknesses of this paper:

I felt there will be questions on what's the "new scientific discovery" here. This paper is mostly a report on empirical insights, that decoupling or reducing the number of adversarial samples does not really hurt performance that much. Most of the gradient recycling discussed in the paper is prior work but with engineering insights.

On experimental results. I take some issues with this "The AdvProp also shows compelling robustness compared with the vanilla training, verifying its effectiveness. However, the comparison is unfair as AdvProp using 7× training budget. In the same budget setting, the performance of 15-epoch AdvProp (dividing the total epochs by 7) degrades significantly." The motivation of the paper is that you can retain the advantage of AdvProp without paying the computational cost, i.e., as the authors called it, a "free lunch". If you can't retain the advantage of advprop, then I question what is the purpose.

**Summary Of The Paper:**

AdvProp is not efficient. This paper tries to overcome that with several measures: (1) decoupling, where they now do not need to have one adversarial sample with each clean image, (2) one step adversarial rather than multi-steps, and (3) reuse recent work on accelerating advprop such as recycling gradient, for which they provide several insights.

**Summary Of The Review:**

I am ambivalent based on what I mentioned above. The paper provides empirical insights, but IMO not huge enough for ICLR, and is well written. However, I am wondering whether there is any new scientific discovery here -- it will be nice for those "tricks" mentioned in the paper, the authors can contrast clearly why they are novel as it seems like they have been previously proposed. I also question the AdvProp vs Fast AdvProp results where we have to significantly reduce AdvProp to have the Fast version beat it. I will be interested to see what the other reviewers think and I am opened to discussions and changing my feedback.

---

> ### Author Response · Authors · 2021-11-22
> **Reply to Reviewer 7oqV**
>
> We first thank the reviewer for the detailed comments and the appreciation of our work. We address the concerns below:
>
> **Q1: Concerns on “new scientific discovery”**
>
> A1: Thanks for raising this concern. As stated in the **general response**, we believe our method is a set of carefully designed strategies for reducing AdvProp’s computational cost, rather than being a bag of tricks. Moreover, we believe our method is able to benefit future research in adversarial learning (e.g., our shuffle BN experiments could be a simple and effective alternative to the well-known robust overfitting issue in adversarial training), and make AdvProp be applicable to different applications and to large-scale settings. We will include these clarifications in the paper.
>
>
> **Q2: Fast AdvProp doesn't retain the advantage of AdvProp**
>
> A2: The key message in this paper is we can leverage adversarial examples to improve recognition models for **free**. As shown in our experiments, without incurring additional training cost, our Fast AdvProp is able to outperform the vanilla baseline using different neural architectures, augmentation strategies and recognition tasks. We will make this concept more clear in our paper.
>
> There is indeed a performance gap between the full-version AdvProp and Fast AdvProp, but we would like to argue such comparison is sort of unfair as the full-version AdvProp is much more computationally expensive than ours. Moreover, as our Fast AdvProp is the first attempt to make AdvProp be as computationally cheap as the vanilla training and meanwhile still yielding better performance than the vanilla training, we believe it is reasonable to leave designing better Fast AdvProp as a future work.

---

### Official Review · Reviewer_63kv · 2021-11-01

**Correctness:** 4
**Technical Novelty And Significance:** 3
**Empirical Novelty And Significance:** 3
**Recommendation:** 8
**Confidence:** 4

**Main Review:**

STRENGTHS:
- The paper is very well written and the used techniques are well explained and each of the modifications to Advprop is justified by ablation experiments.
- The author(s) do(es) a good job at counting the number of forward- and backward propagations to demonstrate the increased performance of Fast Advprop compared to Advprop.
- The overall experimental results are promising. The author(s) show(s) that Fast Advprop is able to improve the ResNet performance on some ImageNet-like datasets while keeping the training budget comparable to the budget for vanilla training.

WEAKNESSES:
- My main criticism is that the proposed Fast AdvProp is a potpourri of existing methods (adversarial propagation (Xie et al., 2020), free adversarial training (Shafahi et al., 2019) and Shuffling BN (He et al., 2020)) and simple modifications (splitting the training data into clean images and ones for which adversarial perturbations are calculated, rebalancing the losses of clean and adversarial images, synchronizing update speed) which limits the novelty of this approach. However, each of the modifications is well justified by ablation studies and the overall experimental evaluation is convincing.
- Table 1 points out the gained/lost accuracy in comparison with the vanilla training for some of the entries. However, for most of the entries this difference is not stated. Is there a reason for this? Why are some of the differences printed in green and not in red? What does the red arrow indicate? This should be explained in the caption of the table.
- Although, the methods used in this paper are well explained and justified, I would have liked some more explanations in Algorithm 1, e.g., which line corresponds to which paragraph. "Fuse the gradients" could for example be expressed as "Rebalancing gradients" to make the connection to the relevant paragraph.

MINOR REMARKS:
- missing word on page 6: "When attack using running statistics and non-targeted attack"
- missing/wrong word on page 6: "Before feed the adversarial images on each GPU into the network"
- page 8: "shuffling BN" vs. "Shuffling BN"
- capitalization of references: "equation 4" vs. "Table 3"


**Summary Of The Paper:**

The paper proposes Fast Advprop which is an modified implementation of adversarial propagation. Adversarial propagation aims at improving the robustness and generalization abilities of deep neural network classifiers by performing adversarial training with additional batch normalization layers that are solely used for the adversarial examples which are used during training. The paper suggests to apply different existing techniques and small modifications to increase the training speed while using AdvProp.

**Summary Of The Review:**

Overall, I would recommend to accept this submission into ICLR 2022. The paper is well-written and the results are good. I would be happy to see some of the points, that I stated in my review, would be implemented in a future version of the paper.

---

> ### Author Response · Authors · 2021-11-22
> **Reply to Reviewer 63kv**
>
> We first thank the reviewer for the detailed comments and the appreciation of our work. We address the concerns below:
>
>
> **Q1: Our method is a collection of existing methods**
>
> A1: Thanks for raising this concern. As we stated in the **general response**, enabling these existing strategies to improve model performance is challenging, and requires a significant amount of non-trivial diagnosis and technical contributions. We will include these clarifications in the paper.
>
>
> **Q2: Table 1 presentation**
>
> A2: Thanks for your suggestion! We will mark all the gained/lost robustness in Table 1.
>
> We use green to indicate performance is dropped compared to the baseline, and red to indicate performance is improved over the baseline. The upward arrow indicates the value is higher the better, and the down arrow indicates the value is lower the better; the color of the arrow does not matter here.
>
> We thank you again for your suggestions and will improve the presentation of Table 1.
>
>
> **Q3: More explanations in Algorithm 1**
>
> A3: Thanks for your suggestions. We will improve the presentation of Algorithm 1 with more explanations.

---

### Official Review · Reviewer_e6SS · 2021-11-02

**Correctness:** 4
**Technical Novelty And Significance:** 4
**Empirical Novelty And Significance:** 4
**Recommendation:** 8
**Confidence:** 5

**Main Review:**

Strength
1. The motivation of the paper is solid and the paper clearly quantifies and compares the training budget between the proposed method and AdvProp.
2. Figure 1 and Algorithm is clear and demonstrates the overall idea of the paper.
3. The paper is well organized and well presented.
4. The proposed method demonstrates better performance under the same computation cost and generalizes to various tasks.
5. The ablation study is thorough and cover different aspects of the proposed method.

Weakness
1. While reading the text, it is unclear what is the rationale of introducing g^{noise} in Algorithm 1. The optimization of g^{noise} could be similar to g^{clean}, but instead operating on X_2 without adding noise (changing the purple block to blue block in Figure 1(b)). What is the performance under such modification? The author is suggested to provide more detail about the introduction of g^{noise}.
2. Table 4 shows the performance of various p_adv value. What is the performance for p_adv < 0.2 ?
3. The robustness for advprop is suggested to add to Table 3 for comparison.


**Summary Of The Paper:**

This paper aims to improve the training speed and decrease the computation cost of AdvProp, which is a method that leverages the adversarial example to improve the image recognition accuracy. AdvProp uses separate batchnorm for clean and adversarial examples respectively. In this work, the proposed method Fast AdvProp reduces the computation cost by reusing some gradient computation during training. In the experiment section, Fast AdvProp demonstrates better image recognition and object detection performance under the same training budget and can be combined with existing training strategies, such as mix up. Overall, this paper proposed an efficient training strategy that can be combined with various existing data augmentation for various tasks.

**Summary Of The Review:**

This paper is enjoyable to read and provides good insight to the problem of large computation cost of AdvProp. The proposed solution Fast AdvProp demonstrates good performance with less computation cost and can potentially be applied to various vision tasks.

---

> ### Author Response · Authors · 2021-11-22
> **Reply to Reviewer e6SS**
>
> We first thank the reviewer for the detailed comments and the appreciation of our work. We address the concerns below:
>
> **Q1: what is $g^{noise}$?**
>
> A1: $g^{noise}$ is introduced by the PGD attacker, whose first step is to generate a random noise ($g^{noise}$) as the initialization of the added adversarial perturbation. As ablated in previous literature [1], this random initialization is the key to stabilizing adversarial training.
>
> We here also ablate its effects on ResNet-50’s Fast AdvProp training:
>
> | Setting   | ImageNet $\uparrow$ |
> | --------- | ------------------- |
> | w/ noise  | 76.53               |
> | w/o noise | 76.46               |
>
> From this table, we can see the effect of this random noise $g^{noise}$ is marginal in AdvProp (i.e., only 0.07% accuracy difference). We will add this interesting result in our paper.
>
>
> **Q2: Influence of $p_{adv}$**
>
> A2: Thanks for your suggestion. We additionally ablate the effect of using a smaller $p_{adv}$.
>
> | Setting        | ImageNet $\uparrow$ | Epochs |
> | ---------        | ------------------- | ------ |
> | $p_{adv}$ = 0.00 | 76.20               | 105    |
> | $p_{adv}$ = 0.11 | 76.49               | 94     |
> | $p_{adv}$ = 0.20 | 76.53               | 87     |
>
> Firstly, we can observe that, by reducing $p_{adv}$ from 0.20 to 0.11, it only marginally decreases the model accuracy by 0.04%. If we further decrease $p_{adv}$ to 0, then our method degenerates to the vanilla training baseline, with 76.20% accuracy.
>
> These results suggest $p_{adv}$ indeed has a sweet point. But meanwhile, its value is fairly robust to a certain range, i.e., our ablations show setting $p_{adv} = 0.11$ or setting $p_{adv} = 0.20$ both yield better results than the vanilla training baseline.
>
>
> **Q3: AdvProp in Table 3**
>
> A3: Thanks for your suggestion. We will add these results (from Table 1) for a clearer presentation.
>
> [1] Wong, Eric, Leslie Rice, and J. Zico Kolter. "Fast is better than free: Revisiting adversarial training." arXiv preprint arXiv:2001.03994 (2020).

---

> > ### Comment · Reviewer_e6SS · 2021-11-29
> > **Post Rebuttal**
> >
> > Thanks the author for providing the rebuttal. While I agree some of the novelty of this paper belongs to the engineering work, it seems to me that speeding up the original AdvProp paper is not so trivial. Moreover, as far as I understand, this method can be applied in most of the classification task, which could benefit the community. After reading the rebuttal and the reviews from other reviewer, I tend to accept the paper and open to discussion. Thanks.

---

### Author Response · Authors · 2021-11-22
**A General Response To Reviewers’ Concerns**

We thank all reviewers for their thoughtful feedback, which will help us improve the quality of this paper. We are delighted to see all reviewers enjoy reading this paper, and appreciate the effectiveness of our method on accelerating AdvProp. The major concern (raised by reviewer 63kv, 7oqV and QrYp) is the novelty of this paper, which we aim to clarify next:

### Technical Contribution

Our Fast AdvProp relies on the following two core designs to reduce the computational cost:

1. Repositioning adversarial examples solely as a bonus part for network training, i.e., the training set for clean samples is not overlapped with the training set for generating adversarial examples

- Though this modification is simple, we argue that **the insight behind it is nontrivial**. Training with both clean images and their adversarial counterparts has been a common protocol for many years. We are the **first** to challenge this training standard, and empirically demonstrate that assigning a small portion of the training set exclusively for generating adversarial examples is enough for yielding promising results. In addition, we believe this simple and effective strategy has the potential to help speed up future research in leveraging adversarial examples.

2. Reusing the gradient when generating adversarial examples

- The gradient-reusing strategy is inspired by [1,2], but, as we stated in Section 3.2.2, naively adopting the existing gradient-reusing strategy [1] collapses the training. Rather than randomly trying different strategies in a brute-force manner, we carefully pinpoint the training instability issue step-by-step and develop the solution accordingly. Therefore, we respectfully disagree that our method is a bag of tricks; rather we believe it is a set of carefully designed strategies to enable gradient reusing to be applicable in AdvProp.

In addition, most of our strategies are the first to be applied in adversarial learning, and have the potential to benefit future research in this direction. E.g., robust overfitting leads to significant instability in fast adversarial training, and is extensively discussed in recent papers [3,4,5]; based on our findings, we believe the shuffle BN strategy [6] may be a simple and effective alternative to resolve this robust overfitting issue.

### Potential Impacts

AdvProp shows that leveraging adversarial examples is a promising way to improve overall model performance. However, the intensive computational cost, mainly due to the high complexity of generating adversarial examples and underlying data redundancy, limits its potential application on large-scale problems (large models like ViT-L, big datasets like JFT-300M, extremely long training epochs) and excludes researchers with limited resources. Our method significantly reduces the computational cost of AdvProp, enabling it to embrace every researcher of interest and to improve model performance under the large-scale setting.

In summary, we believe our novelty contribution is significant (though simple). We will include these clarifications in the paper.

[1] Shafahi, Ali, et al. "Adversarial training for free!." arXiv preprint arXiv:1904.12843 (2019).

[2] Zhang, Dinghuai, et al. "You only propagate once: Accelerating adversarial training via maximal principle." arXiv preprint arXiv:1905.00877 (2019).

[3] Wong, Eric, Leslie Rice, and J. Zico Kolter. "Fast is better than free: Revisiting adversarial training." arXiv preprint arXiv:2001.03994 (2020).

[4] Rice, Leslie, Eric Wong, and Zico Kolter. "Overfitting in adversarially robust deep learning." International Conference on Machine Learning. PMLR, 2020.

[5] Andriushchenko, Maksym, and Nicolas Flammarion. "Understanding and improving fast adversarial training." arXiv preprint arXiv:2007.02617 (2020).

[6] He, Kaiming, et al. "Momentum contrast for unsupervised visual representation learning." Proceedings of the IEEE/CVF Conference on Computer Vision and Pattern Recognition. 2020.

---

### Decision · Program_Chairs · 2022-01-20

**Decision:**

Accept (Poster)

**Comment:**

This paper improves the training speed and decrease the computation cost of AdvProp, which is a method that leverages the adversarial example to improve the image recognition accuracy. The method achieves the speedup by leveraging a collection of practical heuristics, including reusing some gradient computation during training. The paper is well written, well justified with empirical supports, and can be potentially useful in many vision tasks. On the other hand, some novelty of the method is incremental, and the issues regarding empirical results and claims pointed out by the reviewers need  to be addressed in the revision.